# Generation of Chloroplast Molecular Markers to Differentiate *Sophora toromiro* and Its Hybrids as a First Approach to Its Reintroduction in Rapa Nui (Easter Island)

**DOI:** 10.3390/plants10020342

**Published:** 2021-02-10

**Authors:** Ignacio Pezoa, Javier Villacreses, Miguel Rubilar, Carolina Pizarro, María Jesús Galleguillos, Troy Ejsmentewicz, Beatriz Fonseca, Jaime Espejo, Víctor Polanco, Carolina Sánchez

**Affiliations:** 1School of Biotechnology, Universidad Mayor, Santiago 8580745, Chile; ignacio.pezoas@mayor.cl (I.P.); victor.polanco@umayor.cl (V.P.); 2Advanced Genomics Core, Universidad Mayor, Santiago 8580745, Chile; javier.villacreses@mayor.cl (J.V.); caropizarro@genomamayor.com (C.P.); mjgalleguillos@genomamayor.com (M.J.G.); troy@genomamayor.com (T.E.); bfonseca@genomamayor.cl (B.F.); 3Network Biology Laboratory, Centro de Genómica y Bioinformática, Facultad de Ciencias, Universidad Mayor, Santiago 8580745, Chile; 4PhD Program in Integrative Genomics, Universidad Mayor, Santiago 8580745, Chile; miguel.rubilar@mayor.cl; 5National Botanic Garden of Viña del Mar, Valparaíso 2561881, Chile; jespejoc@uc.cl; 6Applied Genomics Laboratory, Centro de Genómica y Bioinformática, Facultad de Ciencias, Universidad Mayor, Santiago 8580745, Chile

**Keywords:** *Sophora toromiro*, chloroplast, genome, SSR, NGS, SNP, molecular markers, conservation, Rapa Nui, Easter Island

## Abstract

*Sophora toromiro* is an endemic tree of Rapa Nui with religious and cultural relevance that despite being extinct in the wild, still persists in botanical gardens and private collections around the world. The authenticity of some toromiro trees has been questioned because the similarities among hybrid lines leads to misclassification of the species. The conservation program of toromiro has the objective of its reinsertion into Rapa Nui, but it requires the exact genotyping and certification of the selected plants in order to efficiently reintroduce the species. In this study, we present for the first time the complete chloroplast genome of *S. toromiro* and four other *Sophora* specimens, which were sequenced de-novo and assembled after mapping the raw reads to a chloroplast database. The length of the chloroplast genomes ranges from 154,239 to 154,473 bp. A total of 130–143 simple sequence repeats (SSR) loci and 577 single nucleotide polymorphisms (SNPs) were identified.

## 1. Introduction

*Sophora toromiro* is an emblematic endemic small tree from Rapa Nui (Easter Island), and even though it had a crucial importance in the history of the island, it has been extinct in the wild since 1960 according to the International Union for Conservation of Nature (IUCN) [1]. Rapa Nui, a tiny land mass with an area of 166 km^2^, suffered the complete transformation of its terrestrial ecology, so that virtually no natural habitat survives [2]. In the beginning, the islanders used toromiro wood to craft ritual totems such as “Moai kava kava” and some other items with high religious importance [3]. The changes of Rapa Nui ecology show that the vegetal resources could not sustain the large growing population, so the islanders slashed and burned the native forest in order to build new houses and grow new crops to feed the people [4]. Also, animal grazing contributed to eradicate the last specimens of the island [5], except the one located inside Rano Kau crater that survived until about 1960 because of its difficult access. Nowadays *S. toromiro* can only be found in botanic gardens around the globe, such as the National Botanic Garden of Viña del Mar in Chile which holds the most well-documented archives and progenies of the last direct *S. toromiro* descendant from the island.

Regarding the germplasm of *S. toromiro*, it has been characterized and referred to as “lines” [6,7]. The results of these studies indicate that there are three large groups: one in Europe, another in Australia and a third in Chile, made up of 4 lines named by the name of their locality or owner. Thus, in Chile, there are currently *S. toromiro* plants in the National Botanical Garden of Viña del Mar (the ones used in this study), Las Brujas Garden (Titze) and Sudzuki. The fourth line named Behn [7] no longer exists. This evidences the possibility that toromiro, when conserved ex situ, may be exposed to pollen from other *Sophora* species that may be growing nearby. In fact, in the literature it is indicated that progeny of the Titze line are the product of hybridizations and perhaps of introgression of cultivated material growing with related species [8]. Indeed, the phenomenon of hybridization or introgression within the Edwardsia section is common and has been reported in both New Zealand and Chile [9,10,11].

The main issue that arises to begin the reintegration of the species into the island is the existence of several hybrid lines of *Sophora*, being the Titze line (*S. toromiro* × *S. microphylla*) [8,12] the most common of the hybrids. *S. microphylla*, introduced from New Zealand into cultivation, and *S. toromiro* are genetically very similar and readily hybridise [13,14]. Previous attempts have been made to reintegrate the species on the island, but the islanders have rejected the trees because they are hybrids, have no certification and do not look like the ancestral specimen. Due to the small number of individuals in existence, the genetic analyzes that have been carried out have not focused on differentiating toromiro from its hybrids [7,15,16] and have used low-resolution techniques with a single molecular marker such as trnH-GUG/psbA [17] or ITS [18]. Therefore, it was decided to carry a genetic study to elucidate the current situation of the existing *S. toromiro* species in the National Botanic Garden of Viña del Mar, and to obtain data to generate a method to determine which trees are offspring of the last Rapa Nui individual and which trees are hybrids, in order to select a population to carry out a plan for the conservation and reintroduction of seedlings of the plants classified as ancestral.

It has been proposed to use methods focused on analyzing the chloroplasts of the collected samples to differentiate species or closely related organisms in a population, using different markers present in their sequences [19,20,21]. Chloroplasts are organelles mainly inherited from the maternal parent [22], which are essential in photosynthesis and other biological processes such as transcription, translation, carbon metabolism, fatty acid synthesis and proteolysis [23,24]. They have a double stranded circular DNA composed by ~130 genes and a total length ranging from 100 to 220 kb [25], structured in a quadripartite way consisting of a large single copy region (LSC) and a small single copy region (SSC) separated by two inverted repeat regions (IRa and IRb) [26,27]. Complete chloroplast genomes and the presence of markers like simple sequence repeats (SSRs) or single nucleotide polymorphisms (SNPs) in their sequences are useful for deciphering phylogenetic relationships between closely related taxa [28].

An SSR, also known as a microsatellite or STR (Short Tandem Repeat), is a sequence motif consisting in 1 to 6 nucleotide units repeated in tandem patterns [29,30]. If the sequence is composed entirely of one repeated motif it is considered a perfect microsatellite, but if it is composed of multiple motifs it is considered a compound microsatellite [31]. Different individuals exhibit quantitative and qualitative variations of tandem repeats thus providing highly informative and reproducible molecular markers. SSRs are enormously useful in studies of population structure, genetic mapping and evolutionary processes and have been the most widely used markers for genotyping plants over the past 25 years [32]. Microsatellites are present in all genomes, with their distribution and location varying between coding and non-coding regions, being more abundant in the latter region [33]. Although these tandem repeats might seem trivial genetic information, they are involved in a variety of regulatory mechanisms including modulation of transcription factor binding and enhancer function, chromatin organization, nucleosome positioning, mRNA stability, cytosine methylation and structural conformation [34].

SNPs on the other hand represents the substitution of a single nucleotide in a position-specific manner within the genome, with applications ranging from resolving a population structure to barcoding generation to correctly differentiate species, among many others [35,36,37]. A major difference between SSRs and SNPs, is that the former can take multiple forms while the latter is mostly bi-allelic, however, the sheer density of SNPs markers that can be found in the genome of any plant or animal, along with is global distribution makes this marker one of the most useful to compare the chloroplast [38,39,40].

For all the reasons mentioned above, we decided to address a high-resolution strategy such as massive parallel sequencing, subsequently generating a bioinformatic pipeline to obtain robust and accurate results as a method to characterize and differentiate the remaining species of *S. toromiro* and its hybrids to help the reintroduction of this symbolic plant to Rapa Nui.

The aim of this study is to find a genetic tool that allows us to differentiate toromiro from its hybrids using genomic information based on the chloroplast sequences, markers distribution and variability inside this organelle.

## 2. Results and Discussion

### 2.1. Chloroplast Database Mapping

Our raw reads were mapped to a reference chloroplast database consisting of 2123 complete chloroplast genomes. The 5 samples had 5.511% of average alignment (2.73 million reads) with the chloroplast database (Table 1), corresponding to the sequences aligned once and more than once. The chloroplast database mapping results showed an alignment percentage within the ~5% expected range [41].

### 2.2. Chloroplast Assembly and Annotation

The reads extracted from the mapping output were assembled with MIRA using the *Sophora alopecuroides* (NC_036102.1) chloroplast genome as reference. The chloroplast assemblies averaged a total depth coverage of 4681x (Table 1). Only one contig was generated in every assembly, thus the N50 value equals the value shown in “Size” (Table 2). Chloroplast genomes consist of a double-strand circular DNA divided in a typical quadripartite structure [42,43] with two inverted repeats (IR) regions, a small single copy region (SSC) and a large simple copy region (LSC) as shown in Table 2, Figure 1 and Figure 2. *Sophora* spp. chloroplasts present a total length of 154,239–154,473 bp (Table 1) consistent with those of *S. alopecuroides* (154,108 bp), *S. tonkinensis* (155.640 bp) and *S. flavescens* (154.378 bp). GC content ranged between 36.43–36.49%. They presented 78 coding sequences (CDS), 32 transfer RNA (tRNA) and 4 ribosimal RNA (rRNA) (Table 2).

We found 17 duplicated genes, marked as “(x2)”, and 15 genes with introns, marked as “^★^“ in Table 3. The duplicated genes are those located in the IR regions, corresponding in order to rpl2, rpl23, trnM-CAU, ycf2, trnL-CAA, ndhB, rps7, rps12, trnV-GAC, rrn16, trnE-UUC, trnA-UGC, rrn23, rrn4.5, rrn5, trnR-ACG and trnN-GUU. The genes with introns are located throughout all the chloroplast genome. Thirteen of them have one intron (atpF, ndhA, ndhB, rpl2, rpoC1, rps12, rps16, trnA-UGC, trnE-UUC, trnK-UUU, trnL-UAA, trnT-CGU and trnV-UAC) and two of them have two introns (clpP and ycf3). These genes are displayed in Figure 1 on a schematic genome map.

We organized the 131 annotated genes by category and functional group which are alphabetically ordered in Table 3. We classified ycf1 and ycf2 as genes with unknown function because there is no certainty about their role, even though some studies show that they may be related to protein imported into the chloroplast [44,45,46]. The assemblies and annotations show similarities in chloroplast genome size, GC content and number of genes compared to those reported in other *Sophora* research [47,48], evidencing the highly conserved nature of the chloroplasts due to its uttermost importance in photosynthesis, transcription, translation, and other important pathways mentioned in Table 3.

The genes shown in Table 3 were organized into a schematic map (Figure 1) according to their real genomic locations. The genes on the outside circle are transcribed counter-clockwise, and those on the inside are transcribed clockwise. A color is assigned to each gene, corresponding to their functional group. The grey inner circle represents the GC percentage and it also delimits the 4 regions of the chloroplast (IRa, IRb, SSC, LSC). These borders are shown in detail in Figure 2, where all 5 samples are aligned according to the quadripartite chloroplast structure. We show the relation of the borders with its closest gene. There are slight base pairs variations, but they are virtually the same.

### 2.3. Phylogenomic Analysis

A multiple sequence alignment of every *Sophora* sample and nine other plants of the *Papilionoideae* subfamily (described in methods) was performed using the complete chloroplast genome sequences. *A. thaliana* was included so that it could be used as an outgroup and a reference to the location of the tree root. The tree was represented as a dendrogram, in which the length of the branches is not representative of the evolutionary distance. 1000 bootstrap repetitions were performed to assess nodal support. The numbers located besides divergences (Figure 3) correspond to the bootstrap support of each clade, which indicate the percentage of bootstrap replicates where the grouping occurs.

It is observed that only 2 clades of the phylogenomic tree have low bootstrap support (<70%, 1000 repetitions), corresponding to the divergence of the hybrid lines. ST1, phenotypically identified as *S. toromiro*, grouped together with ST-hyb1 and ST-hyb2 indicating a great similarity between the chloroplast genomes of these individuals. On the other hand, ST2 diverges closer to the root. These results, together with the phenotypic characterization, suggest that ST2 could correspond to a genuine toromiro. The SM sample, *Sophora macrocarpa,* groups close to the samples that we think could correspond to toromiro (ST2) due to its great similarity and geographic proximity.

Additionally, we aligned the psbA/trnH spacer sequence of our samples with the *Sophora* sequences presented by Shepherd and Heenan [17] in which they found that two genuine *S. toromiro* samples, obtained from herbaria, presented a 3 bp insertion. As expected, four of our samples, ST1, ST2, ST-hyb1 and ST-hyb2, presented the insertion (AAA) in the psbA/trnH spacer excluding SM which belongs to another *Sophora* species (Appendix A). Assuming that maternal inheritance of the chloroplast DNA is occurring in all our samples, the presence of this insertion would indicate that four of them have a *S. toromiro* parent. Considering this, we would have expected to observe less variation between the chloroplast genomes of our four samples that present the 3 bp insertion, because all existing toromiros descend from a very reduced population. We suggest that paternal inheritance [49,50] may be playing a role in the genetic variability exhibited in the chloroplast genomes. These results show that the use of the complete chloroplast genome can be used as a genetic tool to aid in identifying toromiro hybrids because it can identify the chloroplast donor.

### 2.4. Chloroplast Genome Comparison

CgView is a comparative software for circular visualization of genomes that displays information in concentric rings by aligning the queries in a similarity order. The software uses the basic local alignment search tool (BLAST) to perform large-scale comparative analyses, showing the similarities and differences between species. In the outermost part of the circular graph, the genes of the sample used as query (ST2) are indicated, while the subject samples are in the concentric ring from the outside to the inside according to their similarity. Genetic features are mapped into different colors according to its similarities (Figure 4). The black color represents 100% identity between the genomic sequences analyzed. The spectrum of red colors represents identities greater than 90%. The spectrum of blue colors represents identity values of less than 90%. The white color indicates that the sequences compared have 0% identity, so there is no similarity between them.

The ST2 sample is located in the outermost part because it was identified as *S. toromiro* in the phylogenomic analysis shown in Figure 3, and rest of the sequences correspond to the concentric rings ordered from outside to inside as: ST-hyb2, ST1, ST-hyb1, SM, *S. alopecuroides* and *A. thaliana*. Finally, in the innermost part of the graph is represented the GC content and the location of the genome sequence represented in kilo base pairs (kbps). When performing a visual inspection of the map (Figure 4), the great conservation among the genomes is evident, represented by the black colors in the rings, mostly in the IR regions where are rRNAs and tRNAs genes. This high conservation phenomenon goes hand in hand with an increase in GC levels (rRNA ~54% GC and tRNA ~52% GC vs. ~36% GC of the entire chloroplast genome), since it is common for tRNAs to be enriched in high GC palindromic regions that enable the structure to form the characteristic clover shape. The innermost ring, corresponding to *A. thaliana*, presents mostly blue and white colors due to the great differences separating this organism from the *Sophora* genus. Finally, hypervariable regions with identities lower than 82% were identified among the *Sophora* samples, represented by the white lines throughout all the rings, corresponding to sectors of the ycf1 and rpoC2 genes, and the IGS atpI/atpH, petN/psbM, trnE-UUC/psbD and ycf3/psbI.

### 2.5. Phylogenetic Analysis

Hypervariable regions were selected to construct a bootstrap consensus tree, inferred from 1000 replicates, with *S. macrocarpa* (SM) as root. Figure 5 shows the phylogenetic reconstruction of the five *Sophora* samples using the molecular markers ycf1, rpoC2, aptI/atpH, petN/psbM, trnE-UUC/psbD and ycf3/psbI. Nodal support is shown besides divergences. This analysis involved five aligned sequences, one per sample, in which the molecular markers were separately aligned and then concatenated. There were a total of 8084 analyzed nucleotides per sample in the final data set. If chloroplasts are maternally inherited, the ST-hyb2 sample which exhibits phenotypic features of *S. toromiro* and *S. macrocarpa,* should present either a *S. toromiro* chloroplast or a *S. macrocarpa* chloroplast. The same occurs with the ST-hyb1 sample, which exhibits phenotypic features of *S. toromiro* and *S. microphylla*, and should present either a *S. toromiro* chloroplast or a *S. microphylla* chloroplast. The results presented in Figure 5 and Appendix A suggest that four samples present variants of *S. toromiro* chloroplast.

### 2.6. Simple Sequence Repeats (SSR) Analysis

The majority of SSRs are in the intergenic spacers (69.4%) and the large single copy region (72.1%) shown as IGS and LSC, respectively, in Table 4. Mononucleotide SSRs correspond to 76.4% of all the reported tandem repeats. Every sample presented the same compound SSR in the rpoC2/rps2 spacer. We identified 4 unique SSR of ST2 which could be used for future biological validations, corresponding to (TATAT)_3_ and (AAT)_4_ in the matK/rbcL spacer, (AT)_6_ in the atpA/trnR-UCU spacer and (ATTT)_3_ in the rps16 intron. For further information, consult Appendix A.

### 2.7. Single Nucleotide Polymorphism (SNP) Analysis

The whole chloroplast sequences of the 5 samples were aligned for SNP extraction with the aim to find more ways to help in the identification of pure lines of toromiro. A total of 727 single nucleotide polymorphisms were extracted from the multiple sequence alignment (MSA) file using this approach, spread along the whole chloroplast. To filter this data two criteria were used: No SNP with a tri-allelic state and no indel product of gaps in the MSA were allowed. After the filtering, 577 SNPs were kept for the following analyses.

Using the annotated sequences from ST2 as the reference and with a “Bacterial and Plant Plastid” genetic code in SnpEff, the effect and coding sequences variants for all the 577 SNPs were annotated. The distribution of these 577 markers can be seen in the first column of the Appendix A, which covers almost the entire chloroplast. Based on the quadripartite structure of the chloroplast, the IR regions show the less density of effect which in turns shows the fewer SNPs detected by our MSA approach in these regions. A transition/transversion rate (Ts/TV) of 0.524 was found between our samples (Appendix A), and most of the SNPs detected have an effect in the upstream and downstream regions, with 84% of all the detected effects falling into these two using the default setting from SnpEff of 5 kbps of size. Only 4.3% of the effect falls into exonic regions with most of them causing missense mutations (Appendix A). Of all the annotated genes from ST2, 85 have variants generated from the 577 SNPs. The fact that not all the coding genes have variants shows the high conservation of their sequences, so no SNPs were detected for them using our MSA approach. However, this approach shows the full variants generated from all the SNPs and not samples specific or unique SNPs. To do a more specific approach, the SNPs were separated in the 5 samples, filtering them so only SNPs unique to one specific sample were retained. Appendix A shows the resulting distribution of unique SNPs after this new filtering. Almost all the unique SNPs are found in the SM sample, with 303 SNPs assigned to this sample after filtering. This was expected, as this sample corresponds to a different species (*S. macrocarpa*). Among the shared SNP, the sample that has the most SNP in common with SM was ST-hyb2, which is a hybrid between *S. toromiro* × *S. macrocarpa* with 29 shared SNPs. Giving more weight to the phylogenetic analysis from Figure 5, ST1 and ST-hyb1 shared 18 SNPs, more than the 9 SNPs shared between ST1 and ST2, two supposedly pure lines (Table 5).

Using this new subset of SNPs, a new variant analysis was made by comparing the unique markers of ST2, which was identified by our phylogenetic analysis as a pure toromiro, along with ST1 and the hybrid lines ST-hyb1 and ST-hyb2. To help the visualization, regions of 1 kbps without SNPs in any of the four compared samples were left out (Figure 6).

When comparing genes with variants between the 5 samples, the sheer density of markers from the SM sample leaves no room for any of the other 4 samples to show any unique variant genes (Appendix A, upper left diagram). With the aim of finding specific genes that have variants with the specific SNPs in ST2, a comparison of all genes between ST1, ST2 and both hybrids was made. This approach showed 2 genes that have unique variants in ST2, rps7 and ndhB (Appendix A, upper right diagram). When comparing only ST2 and the hybrids, another gene was found, rpoC1 (Appendix A, bottom diagram). Both, rps7, a ribosomal related gene and ndhB, a NADPH-related gene, have duplication in the IR regions with rps7 in the IRa and ndhB in the IRb. At the same time ndhB was detected to have an intron along with rpoC1, a transcription related gene. These three genes along with those with SSR variability could offer a robust marker for *S. toromiro* chloroplast identification.

## 3. Materials and Methods

### 3.1. Samples

Fresh leaves of five geolocalized *Sophora* individuals, identified phenotypically as ST1 *S. toromiro*, ST2 *S. toromiro* (GenBank: MT079958), ST-hyb1 *S. toromiro* × *S. microphylla*, ST-hyb2 *S. toromiro* × *S. macrocarpa* and SM *S. macrocarpa* (GenBank: MT536779), were sampled from Viña del Mar National Botanic Garden, Chile and were stored at −80 °C until DNA extraction.

### 3.2. Genomic DNA Extraction, Quantification and Quality Assessment

Genomic DNA extraction was carried out with GeneAll^®^ Exgene™ Plant SV mini kit using 100 mg of leaflets treated with liquid nitrogen. Quantification was performed with Quant-iT™ PicoGreen^®^ and the purity of the nucleic acid was evaluated by NanoQuant Infinite^®^ 200. The DNA quality was determined by agarose (1%) gel electrophoresis.

### 3.3. DNA Library Preparation and NGS

DNA libraries were generated using the Illumina (San Diego, CA, USA) TruSeq nano DNA kit and they were sequenced with the Illumina HiSeq 2500 platform in the Universidad Mayor Advance Genomics Core and Genoma Mayor. The libraries averaged 50.1 million reads of 150 bp with a Phred Quality Score of 40.

### 3.4. Chloroplast Assembly and Annotation

All raw reads were trimmed and their ids were changed using PRINSEQ v0.20.4 [49]. A chloroplast database was created with Bowtie2 v2.2.3 [50] using 2123 complete chloroplast genome sequences downloaded from National Center for Biotechnology Information (NCBI) Batch Entrez. Our raw reads were mapped to the chloroplast database using Bowtie2. The mapped output was analyzed with SAMtools v1.9 [51], obtaining the reads mapped in proper pair (0 × 2 flag) divided in a forward (0 × 40 flag) and a reverse (0 × 80 flag) strand. The paired end sequences were mapped-assembled with MIRA v4.0.2 [52] using *Sophora alopecuroides* as reference (NC_036102.1). Annotation of chloroplasts genomes was carried out using GeSeq [53]. The IR, SSC and LSC regions were analyzed with IRscope [54]. The schematic diagrams of the circular chloroplast genomes were drawn using OGDraw v1.3.1 [55].

### 3.5. Phylogenomics, Phylogenetics and SSR Search

Besides our 5 *Sophora* samples, we used the complete chloroplast genome sequences of 9 species of the *Papilionoideae* subfamily (NCBI: txid3814), corresponding to *Ammopiptanthus mongolicus* (NC_034742.1), *Ammopiptanthus nanus* (NC_034743.1), *Ormosia hosiei* (NC_039418.1), *Pterocarpus indicus* (MH033831.1), *Pterocarpus tinctorius* (MH033829.1), *Salweenia bouffordiana* (MF449303.1), *Sophora alopecuroides* (NC_045070.1), *Sophora flavescens* (MH748034.1) and *Sophora tonkinensis* (NC_042688.1). Also, *Arabidopsis thaliana* (NC_000932.1) was used as an outgroup. The multiple sequence alignment was made with Clustal-Omega v1.2.4 [56] and the phylogenomic tree was generated with MEGACC v7.0.18 [57] using the maximum likelihood method (ML) and the reverse general time model (GTR + G + I). Bootstrapping was set to 1000 iterations. We compared our five *Sophora* chloroplast genomes with those of *S. alopecuroides* and *A. thaliana* by using CgView Comparison Tool v1.0 (CCT) [58]. Once we identified and selected the hypervariable regions, they were aligned with Clustal-Omega, and then a phylogenetic ML tree was generated with MEGACC, using the Tamura 92 model with Gamma distribution (T92 + G). Bootstrap was set to 1000 iterations as well. The psbA/trnH alignment was made with Clustal-Omega and visualized with JalView v2.11 [59] The SSR search was carried out using MISA v1.0 [60]. The minimum number of repeats were 10, 6, 4, 3, 3 and 3 for the mono, di, tri, tetra, penta and hexanucleotide microsatellites, respectively.

### 3.6. Single Nucleotide Polymorphism (SNP) Search and Characterization

The search for SNPs was carried out using a multiple sequence alignment (MSA) approach at using the whole sequences of the 5 *Sophora* samples using the ClustalW option and default parameters implemented in MEGACC. SNPs were extracted from the MSA file using SNP-sites [61] to generate a VCF file with the full collection of the markers. After filtering and keeping only bi-allelic state markers, the SNPs were annotated using SnpEff [62] using the chloroplast sequence from ST2 as the reference to see the distribution and impact on the genes predicted by GeSeq. A Venn diagram was compiled using the online tool from the Bioinformatics Evolutionary Genomics (www.bioinformatics.psb.ugent.be/webtools/Venn/ (accessed on 15 January 2021)) to select genes uniquely affected by specific SNPs from any of the 5 *Sophora* sequences. Graphics were compiled using GraphPad Prism 7 and adjusted using Illustrator.

## 4. Conclusions

In this study, we show for the first time five complete chloroplast genomes of *Sophora* plants, phenotypically identified as *S. toromiro* (ST1 and ST2), *S. toromiro* × *S. microphylla* (ST-hyb1), *S. toromiro* × *S. macrocarpa* (ST-hyb2) and *S. macrocarpa* (SM). The chloroplast genomes of all the samples were characterized, which adds valuable information to the current knowledge of the *Sophora* genus, with special focus on toromiro. We determined that 4 of our samples (ST1, ST2, ST-hyb1 and ST-hyb2) presented a unique *S. toromiro* molecular marker, reported by Shepherd and Heenan, in the psbA/trnH spacer. SSR were characterized and some unique loci of ST2 were selected to be used in a future biological validation. Using a second approach based on SNPs, 3 genes were found to have variants based on SnpEff analysis which could be used together with the loci detected by SSR analysis to improve the selection and the validation of *Sophora* species. Based on phylogenomics, phenotypic characterization and historical background, we suggest that the ST2 sample could correspond to a genuine *S. toromiro* (non-hybrid), whilst ST1 corresponds to a hybrid line that was phenotypically misclassified. ST-hyb 2, in turn, raises a question about the genetic inheritance of the chloroplasts because its genome presents similarities to both ST2 and SM samples. While ST2 and ST-hyb2 showed the greatest similarities between all the samples in the CCT map, overall, the samples that shared more SNPs correspond to ST-hyb2 and SM. These results suggest that paternal inheritance may be playing a role in the genetic variability of *S. toromiro* chloroplasts. A deeper analysis with a higher number of samples from the National Botanic Garden of Viña del Mar, and the development of nuclear markers is required to give a more accurate picture of the hybridization within the genus, in order to finally achieve a specific barcoding for this species. As a future prospect, the data presented here could help the inhabitants of Rapa Nui to reclaim their legacy with a more precise selection of species.

## Figures and Tables

**Figure 1 plants-10-00342-f001:**
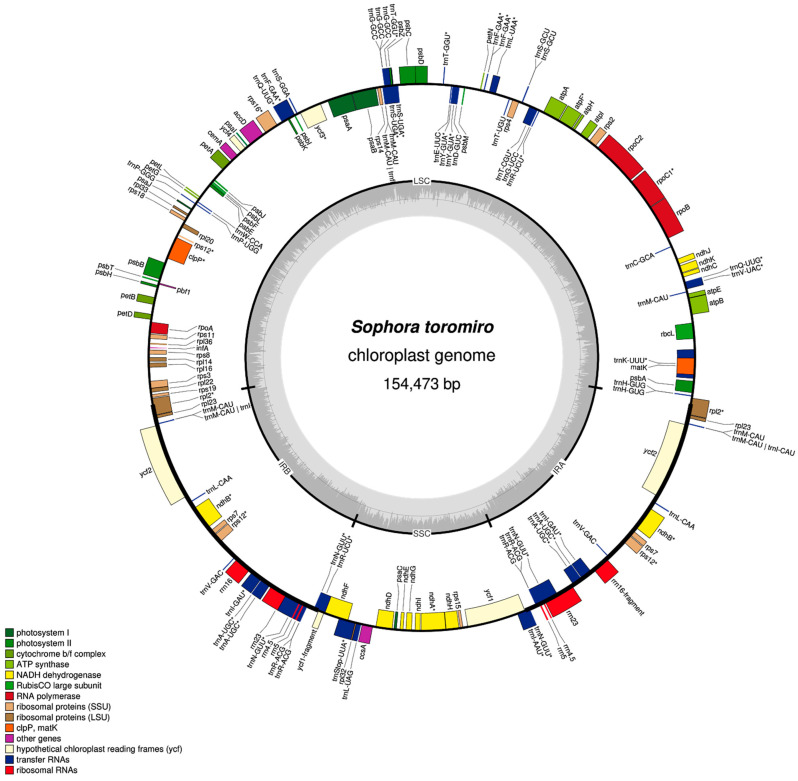
*Sophora toromiro* (ST2) chloroplastidial genome schematic map. Key in colors at bottom left. Genes with introns are indicated with “*”.

**Figure 2 plants-10-00342-f002:**
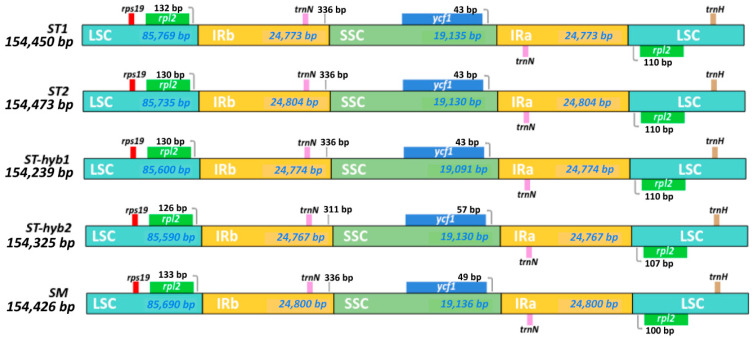
Chloroplast borders comparison between the 5 *Sophora* samples. The length of each segment in base pairs (bp) is shown in blue and the name of the region in white.

**Figure 3 plants-10-00342-f003:**
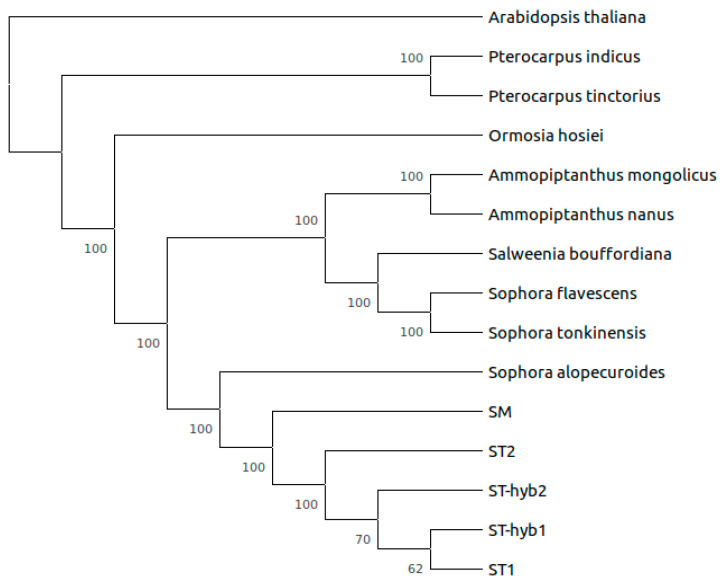
Phylogenomic reconstruction of the relationship of fourteen species from the *Papilionoideae* subfamily with *A. thaliana* as an outgroup.

**Figure 4 plants-10-00342-f004:**
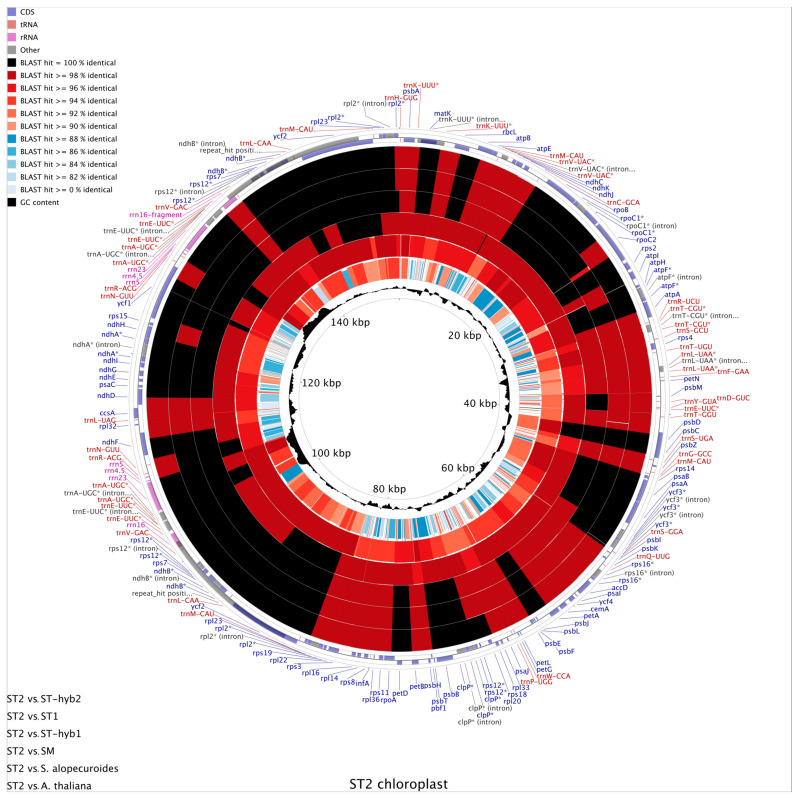
CgView Comparison Tool v1.0 (CCT) map displaying ST2 chloroplast in the outer ring in the form of genes (CDS, rRNA and tRNA). All samples were aligned according to ST2 and were ordered by similarity as shown in bottom left. Blast identity color codes shown in top left. The inner ring shows CG percentage in the ST2 chloroplast genome. Genes with introns are indicated with “*”.

**Figure 5 plants-10-00342-f005:**
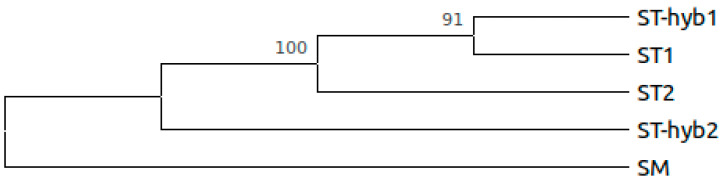
Phylogenetic reconstruction using molecular markers identified through CgView.

**Figure 6 plants-10-00342-f006:**
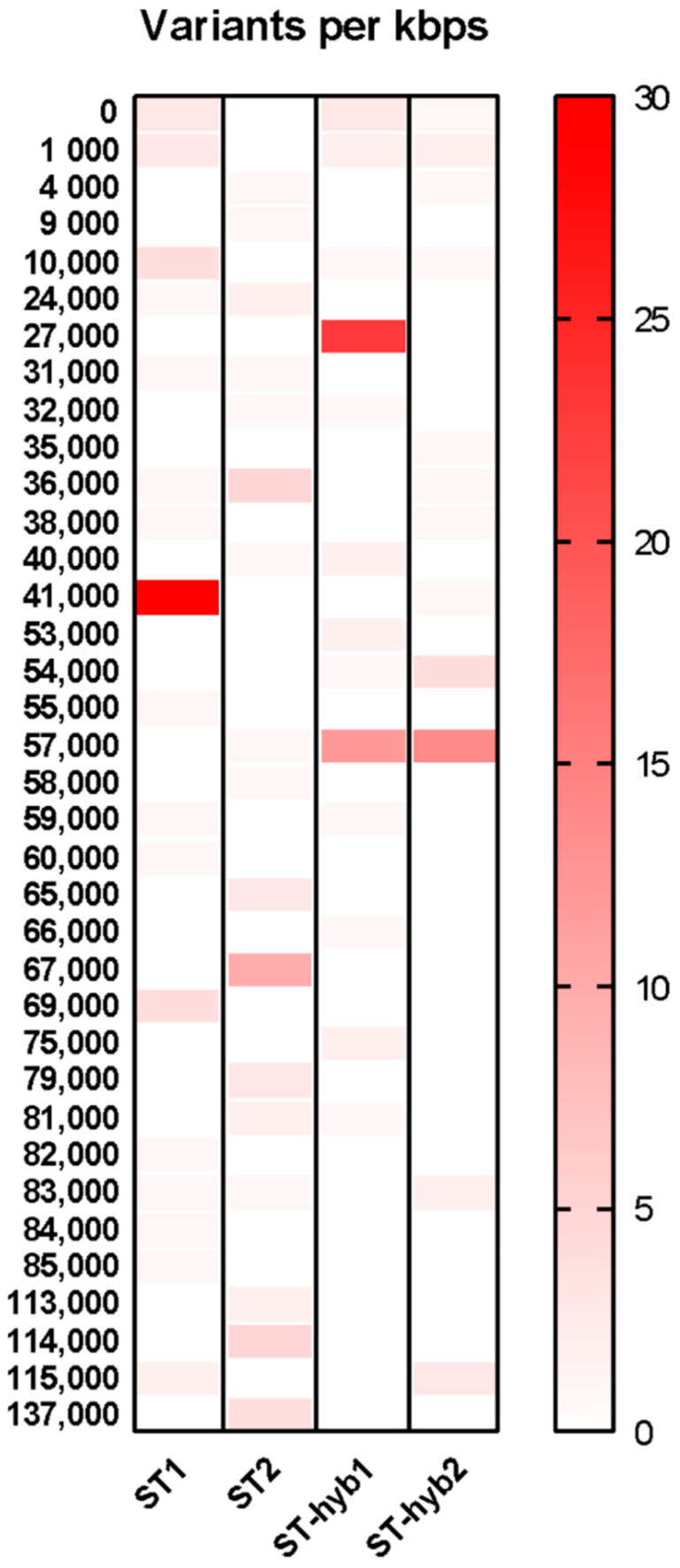
Heatmap of SNP distribution and impact caused in the genes. ST2 annotation data was used as reference for calculating gene variants. Each row represents a window of 1 kbps. Number of variants caused by SNPs are represented with a red coloration to the right.

**Table 1 plants-10-00342-t001:** Alignments, assembly and coverage. The average chloroplast Guanine + Cytosine content (GC) was obtained in the MIRA assembly report.

Species	Raw Data Reads	Alignment Percentage	Aligned Reads	Average Coverage	GC Content	Size (bp)
ST1	48,875,452	4.28%	2,093,045	3549.43	36.48%	154,450
ST2	65,161,387	5.26%	3,424,884	5881.42	36.46%	154,473
ST-hyb1	47,916,209	5.39%	2,581,423	4440.65	36.48%	154,239
ST-hyb2	42,444,815	7.67%	3,253,301	5613.72	36.49%	154,325
SM	46,355,472	4.97%	2,301,647	3920.12	36.43%	154,426

**Table 2 plants-10-00342-t002:** Annotation summary of five *Sophora* complete chloroplasts, indicating their principal features and region length.

Specie	rRNA	tRNA	CDS	Duplicated Genes	Total Genes	LSC (bp)	IR (bp)	SSC (bp)
ST1	4	32	78	17	131	85,769	24,773	19,135
ST2	4	32	78	17	131	85,735	24,804	19,130
ST-hyb1	4	32	78	17	131	85,600	24,774	19,091
ST-hyb2	4	32	78	17	131	85,590	24,767	19,201
SM	4	32	78	17	131	85,690	24,800	19,136

**Table 3 plants-10-00342-t003:** Chloroplast genes ordered according to their categories. Duplicated genes and genes with introns are indicated with (x2) and ^★^ respectively. ATP: adenosine triphosphate; NADPH: nicotinamide adenine dinucleotide phosphate.

Gene Category	Gene Functional Group	Name
Photosynthesis related	ATP synthase	atpA, atpB, atpE, atpF ^★^, atpH, atpI
	Cytochrome b/f complex	petA, petB, petD, petG, petL, petN
	Cytochrome c synthesis	ccsA
	NADPH dehydrogenase	ndhA ^★^, ndhB ^★^, ndhC, ndhD, ndhE, ndhF, ndhG, ndhH, ndhI, ndhJ, ndhK
	Photosystem I	psaA, psaB, psaC, psaI, psaJ
	Photosystem I stability	ycf3 ^★^, ycf4
	Photosystem II	psbA, psbB (x2), psbC, psbD, psbE, psbF, psbH, psbI, psbJ, psbK, psbL, psbM, psbN (pbf1), psbT, psbZ
	Rubisco	rbcL
Transcription and translation related	Ribosomal proteins	rps2, rps3, rps4, rps7 (x2), rps8, rps11, rps12 ^★^ (x2), rps14, rps15, rps16 ^★^, rps18, rps19, rpl2 ^★^ (x2), rpl14, rpl16, rpl20, rpl22, rpl23 (x2), rpl32, rpl33, rpl36
	Transcription	rpoA, rpoB, rpoC1 ^★^, rpoC2
RNA	Ribosomal RNA	rrn4.5 (x2), rrn5 (x2), rrn16 (x2), rrn23 (x2)
	Transfer RNA	trnA-UGC ^★^ (x2), trnC-GCA, trnD-GUC, trnE-UUC ^★^, trnF-GAA, trnfM-CAU, trnG-GCC, trnG-UCC, trnH-GUG, trnI-CAU (x2), trnI-GAU (x2), trnK-UUU ^★^, trnL-CAA (x2), trnL-UAA ^★^, trnL-UAG, trnM-CAU, trnN-GUU (x2), trnP-GGG, trnP-UGG, trnQ-UUG, trnR-ACG (x2), trnR-UCU, trnS-GCU, trnS-GGA, trnS-UGA, trnT-CGU ^★^, trnT-GGU, trnT-UGU, trnV-GAC (x2), trnV-UAC ^★^, trnW-CCA, trnY-GUA
Others	Carbon metabolism	cemA
	Fatty acid synthesis	accD
	Proteolysis	clpP ^★^
	RNA processing	matK
Unknown function	Conserved reading frames	yfc1, ycf2 (x2)

**Table 4 plants-10-00342-t004:** Simple sequence repeats in five *Sophora* chloroplast genomes. #x indicates the repeated nucleotide unit such as 2x = dinucleotide. We included the location (CDS, IGS, intron) and the region (IR, SSC, LSC) of the simple sequence repeat (SSR).

Species	1x	2x	3x	4x	5x	6x	Total	CDS	IGS	Intron	IRa	IRb	SSC	LSC
ST1	100	11	11	7	1	1	131	21	89	21	6	5	24	96
ST2	100	11	12	8	2	1	134	23	90	21	6	6	26	96
ST-hyb1	101	10	11	8	1	1	132	21	92	19	6	5	25	96
ST-hyb2	98	12	11	7	1	1	130	20	91	19	6	6	25	93
SM	113	10	11	7	1	1	143	22	103	18	6	6	29	102

**Table 5 plants-10-00342-t005:** Number of shared single nucleotide polymorphisms (SNPs) between samples.

SNP Shared	ST1	ST2	ST-hyb1	ST-hyb2	SM
ST1	-	9	18	2	8
ST2	9	-	8	2	4
ST-hyb1	18	8	-	4	5
ST-hyb2	2	2	4	-	29
SM	8	4	5	29	-

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
