# Peer review of "Generation of Chloroplast Molecular Markers to Differentiate Sophora toromiro and Its Hybrids as a First Approach to Its Reintroduction in Rapa Nui (Easter Island)"

_plants, 2021, doi:10.3390/plants10020342_

Round 1
Reviewer 1 Report
The manuscript entitled "Generation of chloroplast molecular markers to differentiate Sophora toromiro and its hybrids as a first approach to its reintroduction in Rapa Nui (Easter Island)." was significantly improved after the recommendations.
I do not have any additional comments
Author Response
Dear reviewer
We are glad to have addressed your concerns regarding the previous manuscript.
Even though you don't have additional comments regarding the current manuscript, we outlined below the comments of the other reviewer for you to take into consideration.
- Line 209: we changed herbariums to herbaria as suggested.
- Line 213: we changed parental to parent as suggested.
- Line 346: regarding the identity of the lines used in this study, we address it on lines 348 and 349 (Viña del Mar National Botanic Garden line). All the samples were obtained at this location, thus ST1 and ST2 are from this particular line. We added a comment regarding this on line 55. ST-hyb1, according to the Botanic Garden, is progeny from a Titze-line specimen and at the time of sample collection, it was located in a private greenhouse within the garden. ST-hyb2 was located outside the ST-hyb1 greenhouse, and it presented evident macrocarpa-like phenotypic characteristics that were described by Oscar Fernandez who has been working with toromiro at the Botanic Garden for more than 20 years.
We truly appreciate your comments and suggestions.
Kind regards.
Reviewer 2 Report
This reworking of the manuscript is a big improvement and I feel that my concerns about the original manuscript have been addressed. I only have a couple of minor comments:
Line 209: Change herbariums to herbaria.
Line 213: Change parental to parent.
Line 346: The information in the introduction about the different 'lines' of toromiro is useful. Is it known which lines your ST1 and ST2 samples belong to? This might be good to state here if it is known (researchers may want to sequence the chloroplasts of the other lines that haven't been sequenced yet).
Author Response
Dear reviewer
We are glad to have addressed your concerns regarding the previous manuscript.
Below, we outlined your comments regarding the current manuscript.
- Line 209: we changed herbariums to herbaria as suggested.
- Line 213: we changed parental to parent as suggested.
- Line 346: regarding the identity of the lines used in this study, we address it on lines 348 and 349 (Viña del Mar National Botanic Garden line). All the samples were obtained at this location, thus ST1 and ST2 are from this particular line. We added a comment regarding this in the introduction, on line 55. ST-hyb1, according to the Botanic Garden, is progeny from a Titze-line specimen and at the time of sample collection, it was located in a private greenhouse within the garden. ST-hyb2 was located outside the ST-hyb1 greenhouse, and it presented evident macrocarpa-like phenotypic characteristics that were described by Oscar Fernandez who has been working with toromiro at the Botanic Garden for more than 20 years.
We truly appreciate your comments and suggestions.
Kind regards.
This manuscript is a resubmission of an earlier submission. The following is a list of the peer review reports and author responses from that submission.